# Protection by -Biotics against Hypertension Programmed by Maternal High Fructose Diet: Rectification of Dysregulated Expression of Short-Chain Fatty Acid Receptors in the Hypothalamic Paraventricular Nucleus of Adult Offspring

**DOI:** 10.3390/nu14204306

**Published:** 2022-10-14

**Authors:** Yung-Mei Chao, You-Lin Tain, Wei-Chia Lee, Kay L. H. Wu, Hong-Ren Yu, Julie Y. H. Chan

**Affiliations:** 1Institute for Translational Research in Biomedicine, Kaohsiung Chang Gung Memorial Hospital, Kaohsiung 833, Taiwan; 2Department of Pediatrics, Kaohsiung Chang Gung Memorial Hospital and Chang Gung University College of Medicine, Kaohsiung 833, Taiwan; 3Department of Urology, Kaohsiung Chang Gung Memorial Hospital and Chang Gung University College of Medicine, Kaohsiung 833, Taiwan

**Keywords:** programmed hypertension, maternal high fructose diet, microbial metabolites, short-chain fatty acids, prebiotics, probiotics, synbiotics, butyrate, G-protein coupled receptor 41, olfactory receptor 78, hypothalamic paraventricular nucleus

## Abstract

The role of short-chain fatty acids (SCFAs) in the brain on the developmental programming of hypertension is poorly understood. The present study explored dysregulated tissue levels of SCFAs and expression of SCFA-sensing receptors in the hypothalamic paraventricular nucleus (PVN), a key forebrain region engaged in neural regulation of blood pressure of offspring to maternal high fructose diet (HFD) exposure. We further investigated the engagement of SCFA-sensing receptors in PVN in the beneficial effects of -biotics (prebiotic, probiotic, synbiotic, and postbiotic) on programmed hypertension. Maternal HFD during gestation and lactation significantly reduced circulating butyrate, along with decreased tissue level of butyrate and increased expression of SCFA-sensing receptors, GPR41 and olfr78, and tissue oxidative stress and neuroinflammation in PVN of HFD offspring that were rectified by oral supplement with -biotics. Gene silencing of *GPR41* or *olfr78* mRNA in PVN also protected adult HFD offspring from programmed hypertension and alleviated the induced oxidative stress and inflammation in PVN. In addition, oral supplement with postbiotic butyrate restored tissue butyrate levels, rectified expressions of GPR41 and olfr78 in PVN, and protected against programmed hypertension in adult HFD offspring. These data suggest that alterations in tissue butyrate level, expression of GPR41 and olfr78, and activation of SCFA-sensing receptor-dependent tissue oxidative stress and neuroinflammation in PVN could be novel mechanisms that underlie hypertension programmed by maternal HFD exposure in adult offspring. Furthermore, oral -biotics supplementation may exert beneficial effects on hypertension of developmental origin by targeting dysfunctional SCFA-sensing receptors in PVN to exert antioxidant and anti-inflammatory actions in the brain.

## 1. Introduction

Hypertension is a well-known risk factor for cardiovascular diseases (CVD) [1], the leading cause of mortality and health burden worldwide. Compelling evidence from both animal and human studies suggests that the etiology of hypertension in adults may take origin in early life [2,3], a concept termed developmental origin of health and disease (DOHaD) [4]. Hypertension of developmental origin can be programmed by a number of environmental risk factors [5,6], including those linked to maternal nutrition. Animal and human studies have demonstrated that maternal malnutrition resulting from under- or over-nutrition [7,8], or deficiency in specific nutrients [9,10,11] during gestation and/or lactation, is linked to greater risk for CVD, including hypertension, in later life of the offspring.

In keeping with the DOHaD hypothesis, maternal nutrition also impacts the infant gut microbiome [11,12,13,14,15]. Importantly, since gut microbiota play an active role in the regulation of blood pressure (BP) homeostasis [16,17,18], any change in the composition and/or abundance of the microbiota by maternal nutrition may prime the programming of hypertension in adult offspring. In this context, imbalanced gut microbiota, or microbial dysbiosis, has been implicated in susceptibility to hypertension in offspring to maternal nutritional insults by high fat [19,20], high fructose (HFD) [15,21], or low protein diet [8,22]. Several mechanisms have been put forth to link gut dysbiosis to hypertension [15,16,17,18], of which the dysfunctional gut-brain axis has received considerable attention because of the physiological significance of this axis in various aspects of BP regulation [18,23,24,25]. Evidence supporting a dysfunctional gut-brain axis in hypertension includes perturbed neural trafficking between the gut and autonomic brain regions [24,25,26], increases in total number and activated microglia [27,28], brain oxidative stress [29], and neuroinflammation [28,29,30]; all of which are well-characterized causes of augmented sympathetic drive that leads to elevated BP. The role of gut-brain communication in hypertension of developmental origin, nonetheless, is poorly understood.

The credible notion of gut dysbiosis-induced hypertension prompts the development of nutritional interventions for the prevention and management of hypertension that target gut microbiota. These -biotics include prebiotics (e.g., high-fiber diet), probiotic (e.g., *Lactobacillus* spp.), synbiotics (mixture of prebiotics and probiotics) and postbiotics (e.g., short-chain fatty acids, SCFAs) [11,15]. Gut microbiota-targeted therapies exert beneficial effects against hypertension through the correction of microbial dysbiosis of specific microbial populations and their corresponding metabolites, including the decrease of acetate- and butyrate-producing bacteria and the increase of lactate-producing bacteria [30,31]. Acetate, propionate, and butyrate are the three main components of SCFAs that are formed during bacterial fermentation of non-digestible carbohydrates in the gut and are most widely studied in CVD [32]. SCFAs are natural ligands for a group of orphan G protein-coupled receptors termed free fatty acid receptors (FFARs), including FFAR3 (GPR41) and FFAR2 (GPR43) [33]. SCFAs also activate the olfactory receptor olfr78 [34]. Of note is that in addition to their involvement in energy metabolism, recent evidence suggests that SCFA-sensing receptors play crucial roles in mediating the beneficial effects of SCFAs in BP regulation [31,35,36]. The engagement of FFARs in the remedial actions of SCFAs against programmed hypertension, a subject scarcely studied [37], however, merits further elucidation.

Whereas dysregulated gut-brain axis has been demonstrated in the pathogenesis of hypertension, a perturb relationship between SCFAs and FFARs in the brain and hypertension of developmental origin remains to be demonstrated. Moreover, gut microbiota-targeted therapies against programmed hypertension by the -biotics, including prebiotics, probiotics, synbiotics, and postbiotics, through correction of dysregulated expression of metabolites merit investigation. In particular, it is of interest to explore whether restoration of SCFAs and FFARs signaling in the brain may underlie the beneficial effect of the -biotics on hypertension of developmental origin. Accordingly, the present study aimed to explore the dysregulated tissue levels of SCFAs and expression of FFARs in the hypothalamic paraventricular nucleus (PVN), a major brain region engaged in the regulation of sympathetic activity [38], of offspring to maternal HFD exposure, as well as their rectification that underpins the beneficial effects of prebiotic (fructooligosaccharides, FOS), probiotic (*Lactobacillus gasseri*, *L. gasseri*), synbiotic (FOS+*L. gasseri*) and postbiotic (butyrate) on programmed hypertension. We further investigated whether tissue oxidative stress and neuroinflammation induced by the dysregulated SCFAs-FFARs signaling in the PVN may serve as potential molecular targets of the -biotics in the management of programmed hypertension of developmental origin.

## 2. Materials and Methods

All experiments were performed in accordance with the guideline for animal experimentation as adopted and promulgated by the U.S. National Institutes of Health and were approved by the institutional animal care and use committee of Kaohsiung Chang Gung Memorial Hospital, Taiwan (IACUC no. 2019011803, approved on 22 February 2019).

Virgin female (*n* = 16) and male (*n* = 8) normotensive Sprague-Dawley (SD) rats at the age of 10 weeks purchased from BioLASCO (Taipei, Taiwan) were used. They were housed in animal rooms under temperature-(22 ± 1 °C), humidity-(55 ± 5%) and light-(12:12/light:dark photo-cycle, light on at 08:00) control. Standard laboratory rat chow (PMI Nutrition International, Brentwood, MO, USA) and tap water were available *ad libitum*. All animals were allowed to acclimatize for 14 days in an AAALAC-International accredited animal holding facility in Kaohsiung Chang Gung Memorial Hospital, Taiwan, before experimental manipulations.

### 2.1. Maternal High Fructose Diet Model

A rat model of fetal programming induced by exposure to maternal high fructose diet (HFD) was used in the present study, as previously described [21,37,39]. After mating was confirmed by the observation of vaginal plug (gestation day 0), pregnant female rats were randomly assigned into two groups to receive a normal diet (ND, 46% complex carbohydrate, 3.4 Kcal/g; Harlan Laboratories, Madison, WI, USA) or HFD (60% fructose, 3.6 Kcal/g; TD.89247; Harlan Laboratories) chow during the entire period of pregnancy and lactation. Both food and water were provided *ad libitum*. Epidemiological observations indicate that men are more prone to hypertension at a younger age [40]. Accordingly, only male offspring from litters culled to sizes no more than eight pups after birth were used in subsequent experiments. After weaning (3 weeks after birth), both ND and HFD offspring returned to ND chow until the end of the experiments (at the age of 12 weeks).

### 2.2. Experimental Design

After weaning, the offspring from both ND and HFD dams were housed individually at one rat per cage to avoid cross-contamination of feces. In the first series of experiments, BP was measured under the conscious condition at 4, 6, 8, 10, and 12 weeks (*n* = 8 in each group). Sympathetic vasomotor activity, plasma norepinephrine (NE), and SCFA levels were determined at 12 weeks. Brain tissue containing the PVN was collected at 3, 6, or 12 weeks (*n* = 6 at each time point). Total RNA was isolated for quantification of GRP41, GPR43, and olfr78 mRNA. Total protein was extracted for the detection of acetate, propionate, and butyrate, tissue oxidative stress markers, and proinflammatory cytokines. In some animals (*n* = 6–8), the hypothalamus of rats at 12 weeks that received different treatments was processed for immunohistochemical staining of GPR41, GPR43, and olfr78 immunoreactivity.

In the second series of experiments, a prebiotic, FOS (4 g/kg/day), a probiotic, *L. gasseri* (3 × 10^6^ CFU/mL/day), or a synbiotic (FOS+*L. gasseri*), was supplemented in drinking water to ND and HFD offspring for 4 (from 8–12 weeks of age, *n* = 6–8 per group) or 6 (from 6–12 weeks of age, *n* = 6–8 per group) weeks, or a postbiotic, butyrate (2 g/kg/day, *n* = 7 in each group) for 6 weeks. All measurements performed in the first series of experiments were repeated, and the results from those –biotic-treated ND and/or HFD offspring were analyzed.

In the third series of experiments, gene silencing of GPR41 and olfr78 mRNA was carried out in HFD offspring at 8 weeks (*n* = 5 in each group) by microinjection bilaterally into the PVN of lentiviral vectors encoding short-hairpin ribonucleic acid (shRNA) targeting GPR41 or olfr78s. Effects of the treatments on mRNA expression and distribution of GPR41- or olfr-78-immunoreactive (ir) cells in the PVN, as well as cardiovascular parameters, were examined in HFD offspring at 12 weeks.

### 2.3. Biotic Preparation

The prebiotic supplement FOS (Source Naturals^®^ Inc., Santa Cruz, CA, USA) was prepared by dissolving 2 g into 100 mL of distilled water. Each animal received an average supplement of 4 g/kg/day based on average daily water consumption of 60 mL per animal. The probiotic supplement *L. gasseri* (Swanson Health Products, Fargo, ND, USA) was mixed in distilled water at a concentration of 3 × 10^6^ colony-forming units per milliliter (CFU/mL) and kept refrigerated until use. The postbiotic supplement was prepared by dissolving 5 g sodium butyrate (Sigma-Aldrich, St. Louis, MO, USA) into 500 mL distilled water, which resulted in an average intake of 2 g/kg/day. The doses of prebiotic [37], probiotic [37], synbiotic [41], and postbiotic [42] used were based on previous studies conducted in rats. All drinking water with supplements was changed every 2–3 days, and free access was allowed to rats throughout the experiment.

### 2.4. Blood Pressure Measurements and Spectral Analysis of Systolic Blood Pressure Signals

We routinely measured BP between 14:00 and 16:00 in rats under conscious conditions by the noninvasive tail-cuff method using sphygmomanometry (MK-2000; Momuroki Kikai Co., Tokyo, Japan). Three days before the actual recording sessions, ND and HFD offspring were handled repeatedly and allowed to adapt to the restraint chamber. During the recording sessions, rats were placed in the restraint holder, and the tail was warmed on a warming pad for 10–15 min to increase blood flow for the improvement of data acquisition. A full recording session consisted of five acclimatization cycles to optimize data acquisition, followed by five data acquisition cycles. The averaged systolic BP (SBP) of the five readings was used for statistical analysis. SBP obtained by tail-cuff plethysmography in conscious rats has been validated to be comparable to those measured by radiotelemetry [39].

Additionally, BP recorded directly from a cannulated femoral artery was used to evaluate the low frequency (LF, 0.25–0.8 Hz) component of SBP signals at the age of 12 weeks. Animals used were anesthetized with isoflurane (5% for induction and 2% for maintenance) for cannulation of the femoral artery. BP was recorded for 30 min between 14:00 and 16:00 and was processed by an arterial blood pressure analyzer (APR31a; Notocord, Staffordshire, UK). The digitized SBP signals were subjected to continuous online, real-time power spectral analysis (SPA10; Notocord) based on fast Fourier transform analysis [43]. The power density of the LF component of SBP signals was used as our experimental index to reflect sympathetic vasomotor tone [39].

### 2.5. Measurement of Plasma Norepinephrine

As described previously [44], plasma norepinephrine (NE) level was measured by the *o*-phthaldehyde (OPA) method using high-performance liquid chromatography (HPLC) with fluorescence detection. In brief, the plasma sample was mixed with ice-cold trichloroacetic acid and centrifuged at room temperature, followed by filtering through a syringe filter (0.22 μm; Chroma Technology Corp., Bellows Falls, VT, USA). The sample was then mixed with four-fold methanol, centrifuged again at room temperature, and kept at −80 °C until analyses. The concentration (μg/μL) was computed by comparing the area under the curve of each sample against standard NE solutions of known concentrations. Each sample was analyzed in triplicate, and the mean was used for statistical analysis.

### 2.6. Tissue Collection from the Hypothalamus and Sample Preparation

At the age of 3, 6, or 12 weeks, animals were deeply anesthetized with an overdose of pentobarbital sodium (100 mg/kg, i.p.), followed by intracardial infusion with 500 mL of warm (37 °C) normal saline. The skull was opened, and the forebrain was rapidly removed and immediately frozen on ice. Because of the small size of the PVN in young (3 and 6 weeks) rats, the entire hypothalamic areas between interaural 6.5 to 7.0 mm, based on the atlas of Watson and Paxinos [45], was collected using a rodent brain matrix (World Precision Instruments, Sarasota, FL, USA). In adult (12 weeks) rats, the hypothalamic areas covering the PVN were blocked between interaural 7.0 and 7.5 mm, and PVN tissues surrounding the third ventricle were collected by micropunches. Tissue samples collected from the same experimental groups (*n* = 5–8) were pooled and stored at −80 °C until use.

Total protein extraction: Hypothalamic tissues were homogenized using a Dounce grinder with a tight pestle in an ice-cold lysis buffer that contained a cocktail of protease inhibitors (Sigma-Aldrich) to prevent protein degradation. Solubilized proteins were centrifuged at 20,000 × *g* at 4 °C for 15 min, the supernatant was collected, and total protein was quantified by the Bradford assay with a protein assay kit (Bio-Rad, Hercules, CA, USA).

Total RNA isolation: Total RNA from hypothalamic tissues was isolated with TRIzol reagent (Invitrogen, Carlsbad, CA, USA) according to the manufacturer’s protocol. All RNA isolated was quantified by spectrophotometry, and the optical density 260/280 nm ratio was determined. Reverse transcriptase (RT) reaction was performed using a SuperScript Preamplification System (Invitrogen) for the first-strand cDNA synthesis.

### 2.7. Measurement of Malondialdehyde Level in the PVN

Levels of lipid peroxidation in the PVN were measured by a malondialdehyde (MDA; a primary indicator of lipid peroxidation) assay kit (Biovision, Milpitas, CA, USA), following the protocol provided by the manufacturer. Briefly, PVN samples were reacted with thiobarbituric acid (TBA) at 95 °C for 60 min. The level of MDA-TBA adduct was determined using a microplate spectrophotometer (ThermoFisher Scientific Inc., Waltham, MA, USA), with colorimetric absorbance read at 532 nm [46]. Each sample was analyzed in triplicates, and the mean was used for statistical analyses.

### 2.8. Measurement of Proinflammatory Cytokines in the PVN

The levels of proinflammatory cytokines, including interleukin-1 β (IL-1β), IL-6, and tumor necrosis factor-alpha (TNF-α), in the PVN, were measured using anti-rat ELISA Kits (ThermoFisher Scientific) according to the manufacturer’s specification. The total protein from the PVN was centrifuged for 10 min at 4 °C. The supernatants were used immediately to measure the concentrations of proinflammatory cytokines. Positive and negative controls were included on each plate. The final concentration of the cytokines was calculated by converting the optical density readings against a standard curve. Each sample was analyzed in triplicates, and the mean was used for statistical analyses.

### 2.9. Measurement of Reactive Oxygen Species in the PVN

PVN tissues were homogenized in sodium phosphate buffer (20 mM), centrifuged, and the supernatant was collected for ROS measurement by the electron paramagnetic resonance (EPR) spin trapping technique, as described previously [46]. EPR spectra were captured using a Brucker EMXplus spectrometer (Bruker, Ettlingen, Germany). Typical parameters were set at microwave power: 20 mW, modulation frequency: 100 kHz; modulation amplitude: 2 G; time constant: 655.36 ms; conversion time: 656 ms; sweep time: 335.87 s. We added a membrane-permeable superoxide dismutase (SOD; 350 U/mL) into the incubation medium to determine ROS specificity. Spectra represented the average of 6 scans. Each sample was analyzed in triplicates, and the mean was used for statistical analyses.

### 2.10. Gas Chromatography-Mass Spectrometric (GC-MS) Analysis

Acetate, propionate, and butyrate levels in hypothalamic tissues or plasma were measured using gas chromatography-mass spectrometry (GC2030-QP2020 NX; Shimadzu, Kyoto, Japan) on an Agilent HP-FFAP capillary column (J&W Scientific, Folsom, CA, USA). SCFAs from hypothalamic samples were prepared following previously described procedures [47] with modifications. In brief, 5 mg of pooled hypothalamic sample was vortexed in phosphoric acid (0.5% *v*/*v*) solution for 10 s, followed by homogenization for 4 min at 40 Hz. The samples were then treated with ultrasonic waves for 5 min at 4 °C. After 10 min of centrifugation at 12,000 × *g* at 4 °C, the supernatant was collected for GC-MS analysis. SCFAs in plasma were extracted by adding 0.2 mL of 2-methylvaleric acid as an internal standard to a 0.1 mL plasma sample and vortexed for 30 s. This was followed by centrifugation for 15 min at 12,000 × *g* at 4 °C. The supernatant was kept at −20 °C for 30 min and transferred into a fresh glass vial for GC-MS analysis.

The main conditions of GC-MS analysis were: the front inlet mode was split mode (5:1), the front inlet purge flow rate was 3 mL/min, and the gas flow rate through the column was 1 mL/min. The initial temperature was kept at 80 °C for 1 min, raised to 200 °C at a rate of 10 °C/min for 5 min, then kept for 1 min at 240 °C at a rate of 40 °C/min. The injection, transfer line, quad, and ion source temperatures were 240 °C, 240 °C, 200 °C, and 150 °C. The energy was −70 eV in electron impact mode. The MS data were acquired in Scan/SIM mode with the m/z range of 33–150 after a solvent delay of 3.5 min.

### 2.11. Immunohistochemical Staining

Under deep sodium pentobarbital anesthesia (100 mg/kg, i.p.), rats were perfused intracardially with ice-cold saline, followed by 4% paraformaldehyde in 0.1 M PBS (pH 7.4). The forebrain was harvested and post-fixed in the same fixative, followed by 30% sucrose solution for at least 48 h. 30-μm coronal brain sections were prepared using a freezing microtome (Leica, Houston, TX, USA). The sections were rinsed for 30 min in PBS. After pre-absorption in PBS containing gelatin (0.375%), normal rabbit serum (3%), and triton-X 100 (0.2%), the brain sections were incubated with rabbit polyclonal antibody against rat GPR41 (1:200, #223045; USBiological, Salem, MA, USA), rabbit polyclonal antibody against rat GPR43 (1:200, ABC299; Merck KGaA, Darmstadt, Germany), rabbit polyclonal antibody against rat GRP78 (1:200, #orb228087; Biorbyt Ltd., Cambridge, UK) or rabbit polyclonal antibody against rat anti-ionized calcium binding adaptor molecule-1 (Iba-1; 1:1000, #019-19741; FUJIFILM Wako Chemicals, Richmond, VA, USA) at room temperature overnight and then rinsed three times in PBS. After incubation in biotinylated horse anti-rabbit IgG (1:1000; Jackson ImmunoResearch, West Grove, PA, USA) for 1 h, the sections were rinsed three times in PBS and incubated with AB complexes for 30 min (Vectastain ABC elite kit, Vector Laboratories, Burlingame, CA, USA). Immunoreactivity was detected using EnVision Detection System (Agilent Technologies, Santa Clara, CA, USA, K5007) following the protocol provided by the manufacturer. Sections were mounted and observed using Olympus AX51 bright-field microscope (Olympus Optical, Tokyo, Japan), with the experimenter blinded to the experimental groups. Brain sections of animals subjected to different experimental manipulations were processed together to minimize reaction variations.

### 2.12. Generation of Lentiviral Vector

GPR41 (sc-97148-V, Santa Cruz) and olfr78 (sc-76266-V, Santa Cruz) shRNA lentiviral particles were used in gene silencing experiments. The transduction-ready viral particles contain a target-specific construct that encodes a 19–25 nt (plus hairpin) shRNA designed to knock down gene expression of FPR41 or olfr78. Each vial contains 200 µL frozen stock of 1.0 × 10^6^ infectious units of virus (IFU) in Dulbecco’s Modified Eagle’s Medium with HEPES pH 7.3 (25 mM). Control shRNA lentiviral particles (sc-108080, Santa Cruz) contained a shRNA construct that encodes a scrambled sequence that will not lead to the specific degradation of any known cellular mRNA.

### 2.13. Microinjection of Lentiviral Vectors into the PVN

Microinjection of the lentiviral vectors bilaterally into the PVN was performed according to previously described procedures [46]. Animals were anesthetized with sodium pentobarbital (50 mg/kg, i.p.), placed on a stereotaxic head holder (Kopf, Tujunga, CA, USA), and rested on a thermostatically controlled heating pad. A glass micropipette (external tip diameter: 50–80 μm) connected to a 0.5-μL Hamilton microsyringe (Hamilton Company, Reno, NV, USA) was used to deliver the lentiviral vectors to the PVN at a volume of 50 nL per injection. A total of eight injections (four on each side) of undiluted viral particles (a total volume of 200 nL on each side) were made at two rostrocaudal levels at stereotaxic coordinates of 7.3–7.0 mm posterior to the interaural bar, 0.3–0.6 mm lateral to the midline, and 7.5–8.0 mm below the dorsal surface of the cortex [45]. After the lentivirus injection, the wound was closed in layers, and animals received a postoperative intramuscular injection of procaine penicillin (1000 IU). Animals were allowed to recover in individual cages with free access to food and water.

### 2.14. Quantitative Real-Time Polymerase Chain Reaction

*GPR41* and *olfr78* mRNA levels were analyzed by quantitative polymerase chain reaction (qPCR) using SYBR Green; results were normalized to the GAPDH mRNA signal as described [44]. The following primers were used: *GPR41*: 5′-TTT TCA TGG TGC CCC TGT GT-3′ (forward) and 5′-AAG CCC CAT CAC TCT CTT GC-3′ (reverse); *olfr78*: 5′-ATC AAT GCC CTC TCC CAT GC-3′ (forward) and 5′-TCA GCA CGG CGT TCA GAA TA-3′ (reverse); and *GAPDH*: 5′-AGA CAG CCG CAT CTT CTT GT-3′ (forward), 5′-CTT GCC GTG GGT AGA GTC AT-3′ (reverse). *GPR41* and *olfr78* mRNA were amplified under the following conditions: 95 °C for 3 min, followed by 50 cycles consisting of 95 °C for 10 s, 50 °C for 20 s, 72 °C for 2 s, and finally a 10 min extension at 40 °C. GAPDH was amplified under identical conditions, with the exception of a 55 °C primer annealing temperature. All samples were analyzed in triplicates, and all qPCR reactions were followed by dissociation curve analysis. Relative quantification of gene expression was performed using the 2^ΔΔCT^ method.

### 2.15. Statistical Analysis

All data are presented as mean ± standard deviation (SD). Before all the statistical analyses were performed, the Shapiro–Wilk test was used to confirm that the data complied with normal distribution. Differences in SBP to various treatments were analyzed with a two-way analysis of variance (ANOVA) with repeated measures, followed by Tukey’s multiple comparison test using time and treatment group as the main factors. All other differences in mean values were analyzed by one-way ANOVA, followed by Tukey’s multiple comparison test. Statistical analyses were carried out by employing GraphPad Prism software (version 6.0; GraphPad Software Inc., La Jolla, CA, USA). Values were considered statistically significant at *p* < 0.05.

## 3. Results

### 3.1. Oral Supplement with Prebiotic, Probiotic, or Synbiotic Protects against Programmed Hypertension and Sympathetic Activation in Adult HFD Offspring

We reported previously [21,37,39,44] that compared to ND, maternal HFD exposure during gestation and lactation led to the development of hypertension in adult offspring. Again, we found in the present study that SBP was increased in adult HFD offspring (at the age of 12 weeks) that became significant at six weeks (124.1 ± 8.0 vs. 108.5 ± 10.0 mmHg, *n* = 8, *p* < 0.05). At the same time, plasma NE levels and LF component of SBP signals, our experimental index of sympathetic vasomotor activity [39,43], were elevated in adult offspring (Figure 1). Oral supplement to HFD, but not ND offspring, at 6 or 8 weeks with FOS (4 g/kg/day) and *L.*
*gasser**i* (3 × 10^6^ CFU/mL/day), given alone or together, appreciably protected the adult offspring from programmed hypertension and the increases in plasma NE level and sympathetic outflow (Figure 1). Of note, the observation that protection against programmed hypertension in adult HFD offspring when the supplementation began at 6 or 8 weeks of age suggests that prebiotic, probiotic, or synbiotic supplements are effective in protecting the HFD offspring from programmed hypertension either when implemented at the beginning or after the manifestation of high BP phenotype.

### 3.2. Oral Supplement with Prebiotic, Probiotic, or Synbiotic Alleviates Tissue Oxidative Stress and Inflammation in the PVN of HFD Offspring

Both oxidative tissue stress and inflammation [48,49,50] in the PVN contribute to the neural mechanism of hypertension via the increase of sympathetic outflow. We, therefore, examined whether the protection against programmed hypertension by prebiotic, probiotic, or synbiotic supplements was conferred by targeting those two key mechanisms in the PVN. In the PVN of adult HFD rats, tissue MDA and ROS levels (Figure 2A), as well as IL-1β, IL-6, and TNF-α levels were significantly increased (Figure 2B) when compared to age-matched ND offspring. The distribution of Iba1-ir cells in the PVN of HFD offspring was also increased (Figure 2C). Oral supplements with FOS and *L.*
*gasseri*, given alone or in combination for six weeks, notably diminished oxidative tissue stress and the enhanced expression of proinflammatory cytokines, as well as the distribution of Iba1-ir cells in the PVN of adult HFD offspring. These supplements, on the other hand, exerted no significant effect on tissue oxidative stress and neuroinflammation in the PVN of ND offspring (Appendix A). These results are interpreted to suggest that prebiotic, probiotic, or synbiotic may protect adult HFD offspring from programmed hypertension via their antioxidant and anti-inflammatory actions in the PVN.

### 3.3. Maternal HFD Changes Tissue Butyrate Level and Expression of SCFA-Sensing Receptors in the PVN That Are Restored by Oral Supplement with Prebiotic, Probiotic or Synbiotic

Compelling evidence suggests an association between gut dysbiosis and alterations in their metabolites, such as SCFAs, in the development of hypertension [18,31,34,35,36]. SCFAs modulate BP through the activation of FFARs, including GPR41, GPR43, and olfr78 [33,34], the expression levels of which are subject to modification under hypertensive conditions [29]. In young (3 and 6 weeks) and adult (12 weeks) HFD offspring, tissue levels of butyrate, but not acetate or propionate, were notably decreased (Figure 3), together with an upregulation of GPR41 and olfr78, but not GPR43 mRNA in the PVN (Figure 4). Immunohistochemical staining further revealed differential regional distributions of GRP41 and olfr78 in the PVN of adult offspring. Whereas GPR41-ir cells were mainly located in the lateral magnocellular division of PVN, olfr78-ir cells were distributed in the ventral part of the parvocellular division of PVN (Figure 5). Oral intake of FOS and *L.*
*gasseri**,* alone or together, appreciably reversed the lowered tissue butyrate level (Figure 3), as well as the altered mRNA expression (Figure 4) and distribution of the GPR41- and olfr78-ir cells in the PVN (Figure 5) of adult HFD offspring. The same treatments also restored the reduced plasma butyrate level in HFD offspring (Appendix A). These findings suggest that maternal HFD exposure may perturb gut-brain communication, resulting in a decrease in butyrate level and an increase in GRP41 and olfr78 expression in the PVN of adult HFD offspring. Our data also indicate that such a perturbed SCFA signaling in the PVN could be restored by oral intake of prebiotic, probiotic, and synbiotic.

### 3.4. Gene Silencing of GPR41 and olfr78 in the PVN Alleviates Programmed Hypertension and Sympathetic Activation in HFD Offspring

To interrogate a causal role of GPR41 and olfr78 in the PVN on programmed hypertension in HFD offspring, gene silencing via microinjection bilaterally into the PVN of lentiviral vectors encoding shRNA targeting GPR41 (Lv-GPR41-shRNA; 1 × 10^5^ IFU) or olfr78 (Lv-olfr78-shRNA; 1 × 10^5^ IFU) was performed at the age of 8 weeks in HFD offspring. Transduction of the viral vectors into PVN significantly downregulated the expression of the corresponding mRNA (Figure 6A) and the distributions of GPR41- or olfr78-ir cells (Figure 6B) in the PVN of adult HFD offspring. Compared to scramble shRNA (Lv-sc-RNA) controls, programmed hypertension, increases in plasma NE levels, and LF component of SBP signals in adult HFD offspring were significantly attenuated by microinjection of Lv-GPR41-shRNA or Lv-olfr78-shRNA into the PVN (Figure 6C). These findings are interpreted to suggest the active roles of GPR41 and oflr78 in the PVN on programmed hypertension in adult HFD offspring.

### 3.5. SCFA-Sensing Receptors Participate in Tissue Oxidative Stress and Inflammation in the PVN of HFD Offspring

Our next series of experiments were performed to examine the roles played by GPR41 and olfr78 in tissue oxidative stress and neuroinflammation in the PVN of adult HFD offspring. Compared to Lv-sc-shRNA, microinjection of Lv-GPR41-shRNA or Lv-olfr78-shRNA into the PVN notably alleviated the augmented levels of tissue ROS and MDA (Figure 7A), and the release of proinflammatory IL-1β, IL-6, and TNF-α (Figure 7B), in the PVN of HFD offspring.

### 3.6. Oral Supplement with Butyrate Restores the Expression of SCFA-Sensing Receptors in the PVN and Confers Protection against Tissue Oxidative Stress, Neuroinflammation and Programmed Hypertension in HFD Offspring

Metabolic products of gut microbiota, such as SCFAs, have emerged as important circulating factors in the microbiota-gut-brain axis that participate in the neural mechanism of hypertension [23,24,25,51]. Our final series of experiments were carried out to investigate whether reduced tissue butyrate in the PVN may contribute to the pathogenesis of programmed hypertension in HFD offspring. Compared to vehicle control, oral supplement with postbiotic butyrate (2 g/kg/day) to HFD offspring from the age of 6 weeks restored tissue level of butyrate in the PVN that was comparable to age-matched adult ND offspring (Figure 8A). Furthermore, there was no apparent difference in the expression of GPR41 and olfr78 mRNA in the PVN of butyrate-treated HFD and ND offspring (Figure 8B). The same treatment also protected PVN tissues from oxidative stress and neuroinflammation, as reflected by a reversal of the increased MDA and ROS productions, as well as the elevated releases of IL-1β, IL-6, and TNF-α (Table 1). Oral supplementation with butyrate also significantly diminished programmed hypertension and the elevated plasma NE level and power of the LF component of SBP signals in adult HFD offspring (Figure 8C). These data suggest that a rigorously controlled balance between tissue butyrate levels and expression of SCFA-sensing receptors in the PVN is an essential element in the neural control of BP, which could be the target for prebiotic, probiotic, and symbiotic oral supplementation.

## 4. Discussion

The present study was designed to explore whether a dysfunctional gut-brain axis, particularly the microbial metabolites SCFAs and their receptors in the PVN, causally underpins hypertension programmed by maternal HFD exposure in adult offspring. We also investigated the engagement of SCFA-sensing receptors in the PVN in the beneficial effects of -biotics (prebiotic, probiotic, synbiotic, and postbiotic) on programmed hypertension primed by maternal HFD. Our most noteworthy observations include: (i) demonstration of reduced tissue level of butyrate and increased expression of SCFA-sensing receptors, GPR41 and olfr78, in the PVN of HFD offspring that could be rectified by oral supplement with prebiotic, probiotic, and synbiotic; (ii) maternal HFD-primed programmed hypertension in adult offspring was associated with tissue oxidative stress and neuroinflammation in the PVN and were protected by oral supplement with the -biotics; (iii) gene silencing of *GPR41* or *olfr78* mRNA in the PVN protected against programmed hypertension, as well as oxidative stress and inflammation in the PVN of adult offspring; (iv) oral supplement with the postbiotic butyrate restored tissue butyrate level and expressions of GPR41 and olfr78 in the PVN of HFD offspring; and (v) oral butyrate supplement also protected against programmed hypertension in adult HFD offspring. These data together suggest that alterations in tissue butyrate level and expression of the SCFA-sensing receptors, as well as activation of the SCFA-sensing receptor-dependent tissue oxidative stress and neuroinflammation in the PVN, could be novel mechanisms that underlie hypertension programmed by maternal HFD exposure in adult offspring. Furthermore, oral -biotics supplementation may exert beneficial effects on hypertension of developmental origin by targeting the dysfunctional gut-brain communication in the PVN to exert antioxidant and anti-inflammatory actions.

Dysregulated gut-brain axis in the pathogenesis of hypertension is well recognized [23,24,25], although evidence for the same mechanism in hypertension of developmental origin is wanting. In the present study, we demonstrated, for the first time, a reduction in tissue butyrate level and an increase in SCFA-sensing receptor GPR41 and olfr78 expression in the PVN of HFD offspring, thus providing novel evidence that suggests the machinery to sense SCFAs at this key brain site involved in regulating BP appears to be dysfunctional. We further unveiled that upregulated GPR41 and olfr78 in the PVN may be engaged in hypertension programming in HFD offspring via tissue oxidative stress and neuroinflammation. This notion is supported by data that gene silencing of GPR41 or olfr78 mRNA expression in the PVN ameliorated the augmented tissue levels of MDA and ROS and the elevated release of proinflammatory cytokines of IL-1β, IL-6, and TNF-α in HFD offspring. Tissue oxidative stress and neuroinflammation in the PVN [48,49,50] have been demonstrated to play pivotal roles in the neural mechanism of hypertension via increasing the central sympathetic outflow. Consistent with these demonstrations, we found in this study that microinjection of Lv-GPR41-shRNA or Lv-olfr78-shRNA into the PVN of adult HFD offspring ameliorated the elevated plasma level of NE and sympathetic vasomotor activity programmed by maternal HFD exposure. Interestingly, gene silencing GPR41 and olfr78 in the PVN almost completely protected the hypothalamic tissues from oxidative stress and the release of proinflammatory cytokine (cf. Figure 7) but only partially reduced sympathetic vasomotor tone, plasma NE level, and programmed hypertension (cf. Figure 6C) in adult HFD offspring. These findings suggest that additional brain regions are engaged in the neural mechanism of hypertension of developmental origin. In this regard, we reported recently [39] that upregulation of angiotensin type 1 receptors at the rostral ventrolateral medulla, where sympathetic premotor neurons reside [52], also contributes to sympathetic overactivation and hypertension programming of offspring to maternal HFD exposure.

The most well-studied function of GPR41 in the central nervous system (CNS) is the activation of sympathetic ganglionic neurons and the release of NE by propionate [53]. The present study expands the function of GRP41 in the CNS to suggest that the PVN is an additional active brain region through which this FFAR may regulate sympathetic vasomotor tone and NE release, hence neural mechanism in BP regulation. On activation, GPR41 is coupled to G_i/o_ to inhibit cAMP production and promotes extracellular signal-regulated kinase1/2 (ERK1/2) phosphorylation [54]. Activation of the ERK1/2 signal and its downstream pathways are known to mediate the production of ROS and proinflammatory cytokines in the CNS [55,56]. At the same time, increased ERK1/2 signaling in the PVN has been reported in DOCA-salt-induced hypertension [57] and proinflammatory cytokine-induced hypertension [50]. Whether the same molecular signaling is engaged in GPR41-dependent tissue oxidative stress and neuroinflammation in the PVN, leading to programmed hypertension in HFD offspring, nonetheless, remains to be elucidated. We are aware that our finding of an upregulation of GPR41 mRNA in the PVN of HFD offspring is opposite to a downregulation of the same FFAR in the PVN of spontaneously hypertensive rats (SHR) [29,58]. The reason for this discrepancy is not immediately clear but might be related to reports that different models of hypertension exhibit different gut microbial dysbiosis [21,37,58,59]. In addition, whether gut microbial dysbiosis and alterations in microbial metabolites are a cause or a consequence of hypertension is still indecisive [60]. Accordingly, the discrepancy could be related to hypertension staging during which GPR41 expression is examined. In contrast to studies in adult SHR with established hypertension, we found in the present study that upregulation of GPR41 mRNA in the PVN was detected in offspring at an early age (3 weeks) before hypertension phenotype was observed in young (6 weeks) HFD offspring.

Olfr78 is an olfactory receptor that was identified as an SCFA receptor with greater preferences for acetate and propionate than butyrate [34]. Most of our knowledge on the role of olfr78 in BP regulation is from olfr78 null mice. The exhibition of lowered plasma renin levels and BP [61] implicates a systemic renin-mediated hypertensive effect on the activation of olfr78 in those mice. Our observations provide new evidence for a hitherto uncharacterized role for olfr78 in the PVN in the neural mechanism of hypertension of developmental origin via ROS production, neuroinflammation, and an increase in sympathetic outflow. These findings are in agreement with observations that gene expression of olfr59 (the rat ortholog of olfr78) is increased in the PVN of SHR and is associated with oxidative tissue stress, greater proinflammatory cytokine release, and increased sympathetic activity [29]. They, however, are opposite to another study that reported reduced expression of *o**lfr59* mRNA in the hypothalamus of SHR [58].

An increase in the number of activated microglia in the PVN plays a pivotal role in the development and maintenance of hypertension [28]. We found in this work an increase in Iba1-ir in the PVN of HFD offspring. Interestingly, olfr78-ir has been found to be distributed in mouse hypothalamic parenchyma that exhibit Iba1-ir [62], providing cytoarchitectural evidence for the potential participation of olfr78 in the release of proinflammatory cytokines in the PVN of HFD offspring. Further experiments are needed to prove this hypothesis. In addition to GPR41 and olfr78, GPR43 is another FFAR that plays an active role in BP homeostasis by exerting a hypotensive effect [31,36,61] and whose expression level in the PVN is modified by hypertension [31,36]. Nonetheless, we found no significant change in GPR43 mRNA in the PVN of HFD offspring at the age of 3, 6, or 12 weeks, or its distribution in the PVN of adult offspring, implying that its role in the pathogenesis of hypertension programmed by maternal diet might not be as impactful as the other two SCFA-sensing receptors in the PVN.

Higher BP level is associated with lower plasma butyrate level in hypertensive animals [17,18,31] and patients [63]. In the present study, we found the plasma level of butyrate is lower in the juvenile HFD offspring and remained low into adulthood despite the fact that they were weaned onto an ND at 3 weeks of age. These findings are in agreement with previous reports [17,18,31,63] and support the notion that maternal diet may play a significant role in shaping the transmission of commensal microbiota that can persist beyond infancy and may extend into adulthood [12], observations that reverberate the DOHaD hypothesis. Although we did not analyze commensal gut microbiota, microbial dysbiosis characterized by reduced butyrate-producing bacteria, such as Akkermansia, Bacteroides, and Lactobacillus, have been reported in young offspring programmed by maternal diet [20,21,37]. As butyrate is able to enter circulation and cross the blood-brain barrier [64] via specific butyrate transporters [65], we speculate that gut butyrate may thus have direct effects on brain regions that are involved in regulating BP. Indeed, our data showed that the butyrate level in the PVN of HFD offspring is reduced when compared to ND offspring and that oral butyrate supplement that effectively restored hypothalamic butyrate levels also protected the offspring from hypertension programmed by maternal HFD. We also observed that tissue oxidative stress and neuroinflammation in the PVN of HFD offspring were alleviated by oral supplement with butyrate. Together, these data suggest that lower availability of butyrate in brain tissue coupled with butyrate-reversible tissue oxidative stress and neuroinflammation in the PVN may contribute to programmed hypertension in HFD offspring. Chronic oral butyrate administration prevents the onset of hypertension in SHR [66]. Mechanisms underlying the antioxidant and anti-inflammatory action of brain butyrate in programmed hypertension, however, require further investigation. In this regard, butyrate activates nuclear transcription factors, such as peroxisome proliferator-activated receptor γ [67], to antagonize nuclear factor-κB signal transduction and exert an anti-inflammatory effect in the gut. It also activates the ATP-activated protein kinase signaling pathway to alleviate oxidative stress-induced intestinal epithelium barrier injury [68].

The credible notion of gut microbial dysbiosis in hypertension prompts the design of gut microbiota modulation as a treatment modality for hypertension. Given its reported beneficial effect on BP homeostasis, and our observation of reduced circulatory and PVN levels of butyrate in HFD offspring with programmed hypertension, we further focused on bacteriotherapeutic supplementation previously demonstrated to increase butyrate production and/or proliferation of butyrate-producing microbes to explore their beneficial effects on programmed hypertension. We found that oral supplement with prebiotic FOS and probiotic *L. gasseri*, given alone or in combination (synbiotic), significantly increased the level of butyrate in plasma and PVN of HFD offspring. The same treatments also restored GPR41 and olfr78 expression in the PVN to levels that were comparable to ND offspring, suggesting that dysfunctional gut-brain axis at the level of PVN could be a common denominator on which all the –biotic therapies target for their beneficial effects on programmed hypertension in adult HFD offspring.

FOS is one of the well-commercialized and investigated prebiotics that is selectively consumed by probiotic bacteria (e.g., Bifidobacteria) to increase butyrate production and/or proliferation of butyrate-producing microbes in healthy adult humans and animals [69]. Although it may vary within specific host contexts, probiotics containing lactic acid bacteria (e.g., *Lactobacilli* spp.) are generally regarded as the “good” bacteria by increasing butyrate production and are commonly used in probiotic supplements [70,71]. Studies performed in hypertensive humans and animal models of hypertension have reported that supplementation with different prebiotic, probiotic, or synbiotic agents enhances acetate, butyrate, and propionate production, leading to the amelioration of hypertension [72,73]. In addition, increases in acetic, butyric, and propionic acids in plasma have been demonstrated after ingestion of FOS in different animal models, including SHR [74,75]. At the same time, long-term administration of *L*. *coryniformis* plus *L. gasseri* reduces BP and decreases vascular inflammation in SHR [76]. Furthermore, systematic reviews and meta-analyses of clinical trials showed reductions in SBP in hypertensive patients supplemented with different synbiotic formulations that contain FOS and different Lactobacillus, Streptococcus, and Bifidobacterium strains [77,78]. It is interesting to note in the present study that all the beneficial effects of the -biotics are comparable regardless of whether they were given alone or in combination, suggesting the therapies may target the same signals (i.e., butyrate/SCFA-sensing receptors in the PVN) for their actions. It is also noteworthy that the protection against programmed hypertension in HFD offspring was observed when the therapies were implemented at the stage when hypertension is under development (at the age of 6 weeks) or when high BP is established (at the age of 8 weeks). This implies that the therapies might aim to prevent and manage hypertension of developmental origin. These findings are in concert with previous studies [15,37] showing that supplementation with probiotics or prebiotics to the mother during pregnancy and lactation prevents adult offspring from hypertension programmed by maternal nutritional insults.

The DOHaD hypothesis of disease pathogenesis canvases a captivating opportunity for disease prevention by implementing interventions from adulthood to young age when the disease is in the asymptomatic phase [20]. To our knowledge, this is the first study that indicates that correction of dysfunctional gut-brain communication may account for the beneficial effects of -biotics in young offspring on hypertension of developmental origin. Nonetheless, we recognize that there are several limitations to our study. First, the amount of -biotics in drinking water for each animal could not be normalized. This precludes us from comparing the therapeutic efficacies and treatment durations of each -biotics. Second, to adhere to animal 3R regulation required by our IACUC, we have to restrict the bacteriotherapeutic supplementation only to HFD offspring in some experiments. Therefore, we are unable to decipher whether the therapies exert differential modification on gut microbiota profile and hence microbial metabolites between HFD- and ND-treated offspring. Third, probiotic and prebiotic therapies have been shown to exert similar BP-lowering effects but have different mechanisms for the modulation of gut microbiota [37]. Exclusion from experimental design to analyze gut microbiota is, therefore, another limitation. Fourth, in this study, the effects of -biotics on SCFAs were only examined for butyrate mainly because it showed the most prominent change in the plasma of HFD offspring. It remains to be determined if the -biotics would affect other SCFAs and/or the FFARs in the PVN for the observed beneficial effects, considering an overlapping in bacterial populations for the production of different SCFAs. In this regard, maternal supplementation with prebiotic inulin increases propionate, whereas probiotic *L. casei* decreases acetate in the plasma of offspring from dams exposed to HFD [37]. The same treatments, however, have no effect on the reduced GPR41 expression in the kidney of HFD offspring.

Our findings, furthermore, raise important questions that warrant future investigations. We found that the levels of acetate and propionate, the natural ligands for GPR41 and olfr78, did not change in plasma or the PVN of HFD offspring. Accordingly, the mechanism underlying the upregulation of their corresponding receptors in the PVN of adult HFD offspring remains to be identified. In this regard, butyrate functions as a histone deacetylase inhibitor [79,80], and epigenetic gene regulation has been proposed in hypertension programmed by maternal diet [11,20]. As such, it would be interesting to explore the role of butyrate in the PVN in the epigenetic regulation of genes encoding the GPR41 and olfr78. In addition, lactate has been reported to be an integral ligand for olfr78 [81]. Therefore, it will also be of interest to examine changes in lactate-producing bacteria and lactate levels in plasma and the PVN of the HFD offspring, as well as the responses of these changes to bacteriotherapeutic supplementation.

## 5. Conclusions

In conclusion, our results demonstrate that oral supplements to young offspring with -biotics, even after the onset of BP phenotype, could protect hypertension development in adult offspring subjected to maternal HFD exposure. The beneficial effects of the -biotics are mediated by rectifying the dysfunctional gut-brain communication via targeting the microbial metabolite butyrate and the SCFA-sensing receptors GPR41 and olfr78, as well as SCFA-sensing receptor-dependent tissue oxidative stress and neuroinflammation in the PVN. These data not only reinforce the hypothesis of a dysfunctional gut-brain axis in the development and maintenance of hypertension but also provide the impetus to consider nutritional interventions as a promising non-pharmacological approach for the prevention and management of hypertension of developmental origin.

## Figures and Tables

**Figure 1 nutrients-14-04306-f001:**
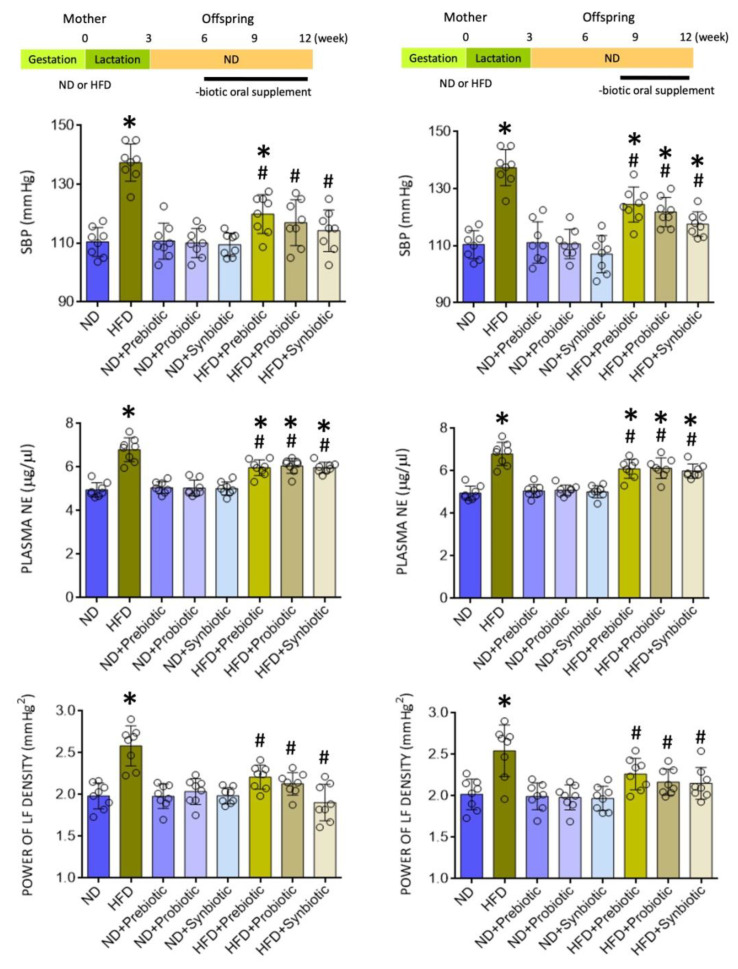
Systolic blood pressure (SBP), plasma norepinephrine (NE) levels, and power density of the low frequency (LF) component of SBP signals in adult offspring from mothers exposed to high fructose diet (HFD) or normal diet (ND) that received oral supplementation of FOS (4 g/kg/day, probiotic), *L. gasser**i* (3 × 10^6^ CFU/mL/day, probiotic) or FOS+*L. gasseri* (synbiotic) initiated at the age of 6 (**left panels**) or 8 weeks **(right panels**). Also presented are schematic diagrams of the experimental design. Data are presented as mean ± SD, *n* = 8 in each group. * *p* < 0.05 vs. ND group, and ^#^
*p* < 0.05 vs. HFD group in the post hoc Tukey’s multiple-range test.

**Figure 2 nutrients-14-04306-f002:**
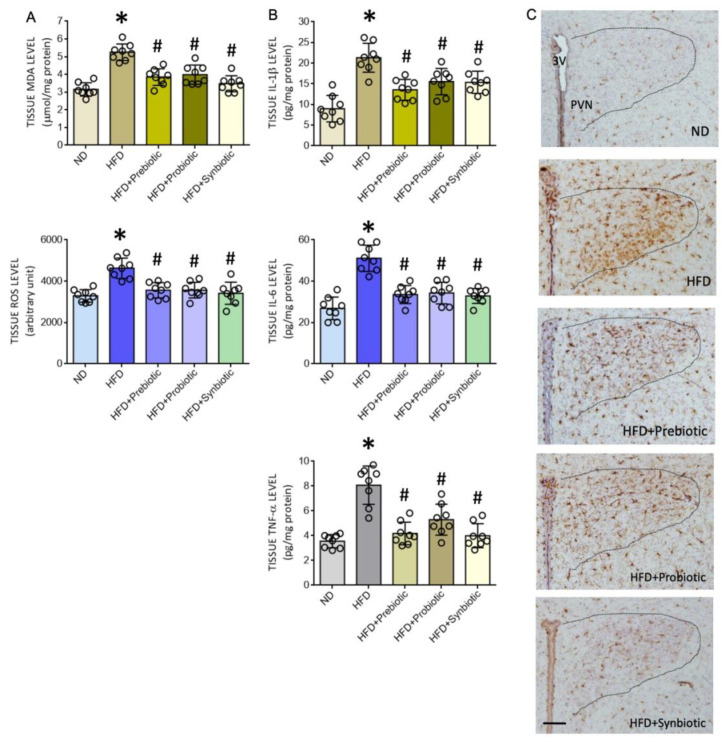
Tissue malondialdehyde (MDA) or reactive oxygen species (ROS) levels (**A**), levels of proinflammatory cytokines interleukin 1-β (IL-1β), interleukin 6 (IL-6), and tumor necrosis factor α (TNF-α) (**B**), and representative photomicrographs showing the distribution of Iba1-immunoreactive cells (**C**) in the hypothalamic paraventricular nucleus (PVN) of adult ND or HFD offspring that received oral supplementation of FOS (4 g/kg/day, probiotic), *L.*
*gasser**i* (3 × 10^6^ CFU/mL/day, probiotic) or FOS+*L. gasseri* (synbiotic) was initiated at the age of 6 weeks. Data are presented as mean ± SD, *n* = 8 in each group. * *p* < 0.05 vs. ND group, and ^#^
*p* < 0.05 vs. HFD group in the post hoc Tukey’s multiple-range test. 3 V, third ventricle; PVN, hypothalamic paraventricular nucleus. Scale bar in (**C**): 200 μm.

**Figure 3 nutrients-14-04306-f003:**
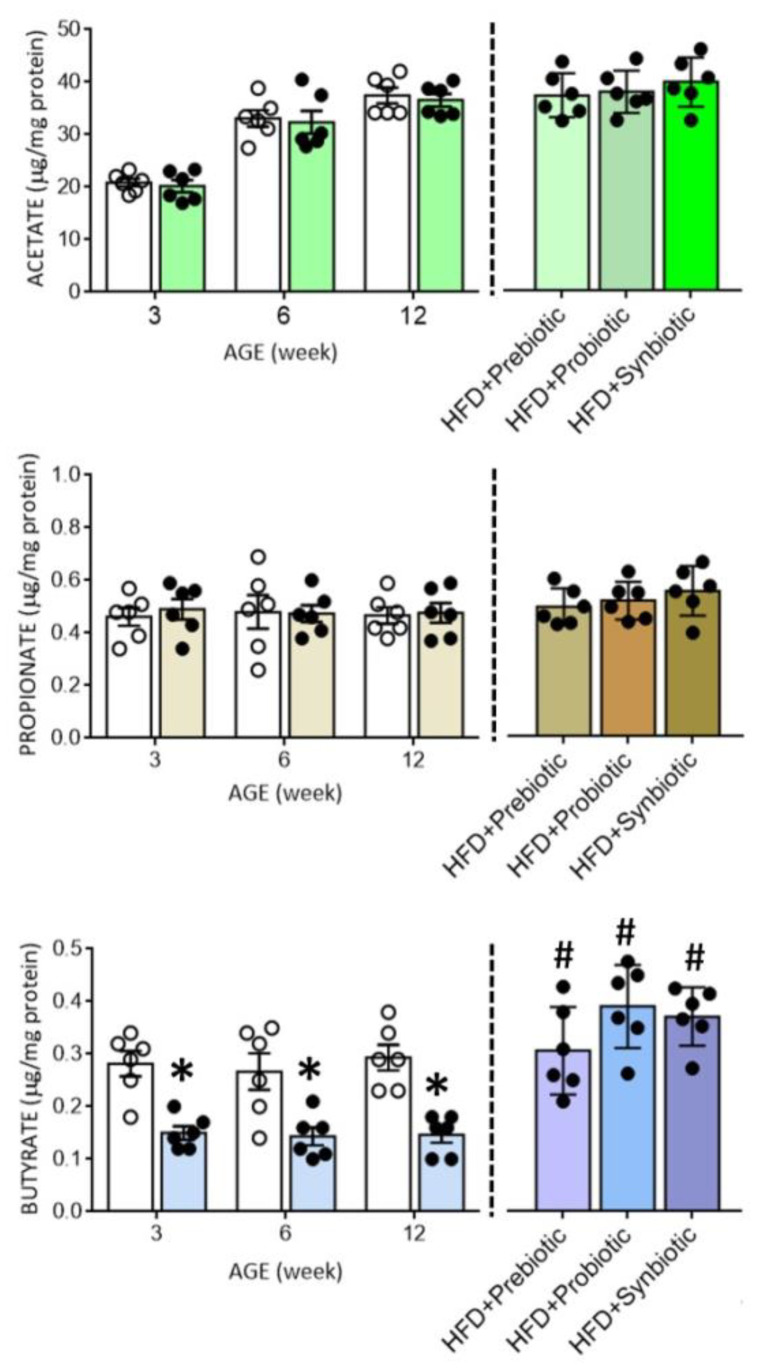
Tissue levels of acetate, propionate, or butyrate in the hypothalamic paraventricular nucleus (PVN) of ND (open circles) or HFD (filled circles) offspring at the age of 3, 6, and 12 weeks or at 12 weeks after additional oral supplementation of FOS (4 g/kg/day, probiotic), *L.*
*gasser**i* (3 × 10^6^ CFU/mL/day, probiotic) or FOS+*L. gasseri* (synbiotic) was initiated at the age of 6 weeks. Data are presented as mean ± SD, *n* = 6 in each group. * *p* < 0.05 vs. ND group, and ^#^
*p* < 0.05 vs. HFD group in the post hoc Tukey’s multiple-range test.

**Figure 4 nutrients-14-04306-f004:**
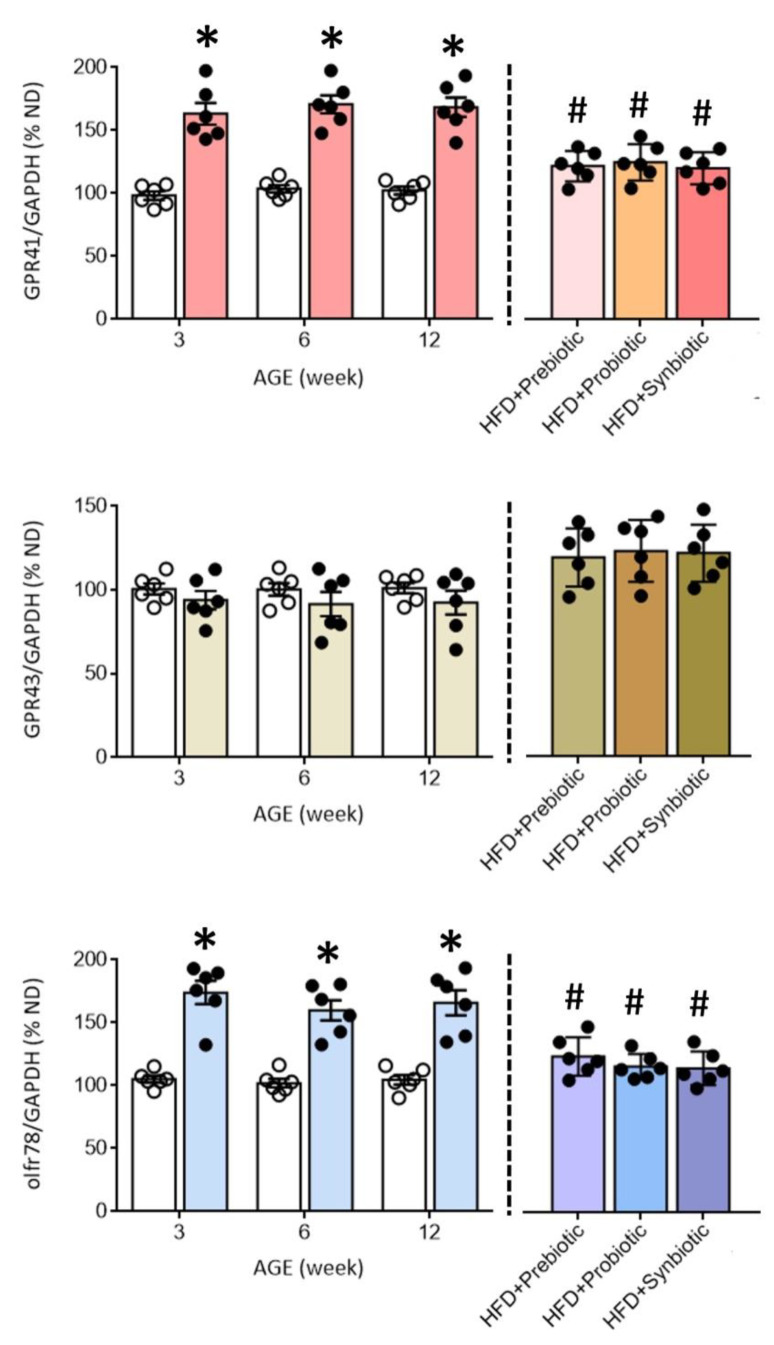
Percentage change in the expression of GPR41, GPR43, and olfr78 mRNA in the PVN of ND (open circles) or HFD (filled circles) offspring at the age of 3, 6, and 12 weeks or at 12 weeks after additional oral supplementation of FOS (4 g/kg/day, probiotic), *L. gasseri* (3 × 10^6^ CFU/mL/day, probiotic) or FOS+*L. gasseri* (synbiotic) was initiated at the age of 6 weeks. Data are presented as mean ± SD, *n* = 6 in each group. * *p* < 0.05 vs. ND group, and ^#^
*p* < 0.05 vs. HFD group in the post hoc Tukey’s multiple-range test.

**Figure 5 nutrients-14-04306-f005:**
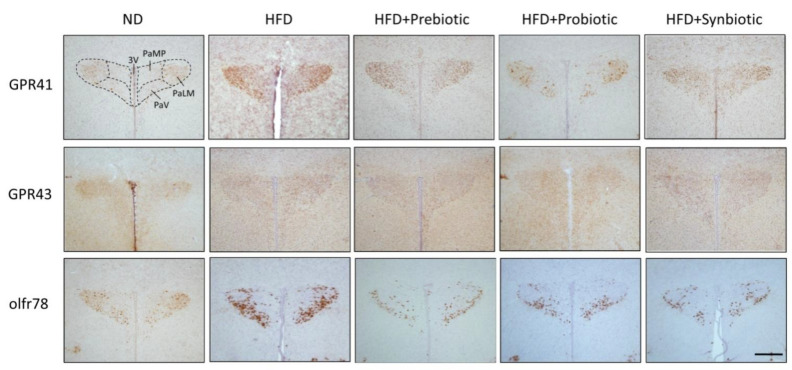
Representative photomicrographs showing the distribution of GPR41-, GRP43- or olfr78-immunoreactive cells (brown color) in the PVN of adult ND or HFD offspring, alone or with additional oral supplementation of FOS (4 g/kg/day, probiotic), *L. gasseri* (3 × 10^6^ CFU/mL/day, probiotic) or FOS+*L. gasseri* (synbiotic) was initiated at the age of 6 weeks. 3V, third ventricle; PaMP, paraventricular nucleus, median pavicellularis; PaLM, paraventricular nucleus, lateral magnocellularis; PaV, paraventricular nucleus, ventrolaris. Scale bar: 500 μm.

**Figure 6 nutrients-14-04306-f006:**
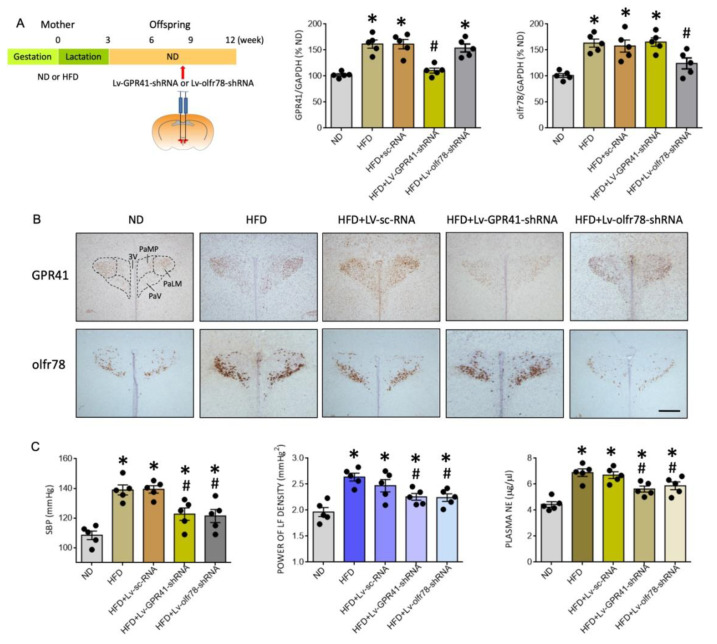
Percentage change in the expression of GPR41 and olfr78 mRNA (**A**), representative photomicrographs of GPR41- or olfr78-immunoreactive cells (brown color) in the PVN (**B**), and SBP, plasma NE levels and power density of LF component of SBP signals (**C**) in adult ND or HFD offspring, alone or with bilateral microinjection into the PVN of lentiviral vectors encoding shRNA targeting GPR41 (Lv-GPR41-shRNA; 1 × 10^5^ IFU), olfr78 (Lv-olfr78-shRNA; 1 × 10^5^ IFU) or scramble shRNA (Lv-sc-RNA) control, performed at the age of 8 weeks. Also presented in (**A**) is the schematic diagram of the experimental design. Arrow indicates the age (8 weeks) when the lentiviral vector is microinjected into the RVLM. Data are presented as mean ± SD, *n* = 5 in each group. * *p* < 0.05 vs. ND group, and ^#^
*p* < 0.05 vs. HFD group in the post hoc Tukey’s multiple-range test. 3V, third ventricle; PaLM, paraventricular nucleus, lateral magnocellularis; PaMP, paraventricular nucleus median pavicellularis; PaV, paraventricular nucleus ventrolaris. Scale bar in (**B**): 500 μm.

**Figure 7 nutrients-14-04306-f007:**
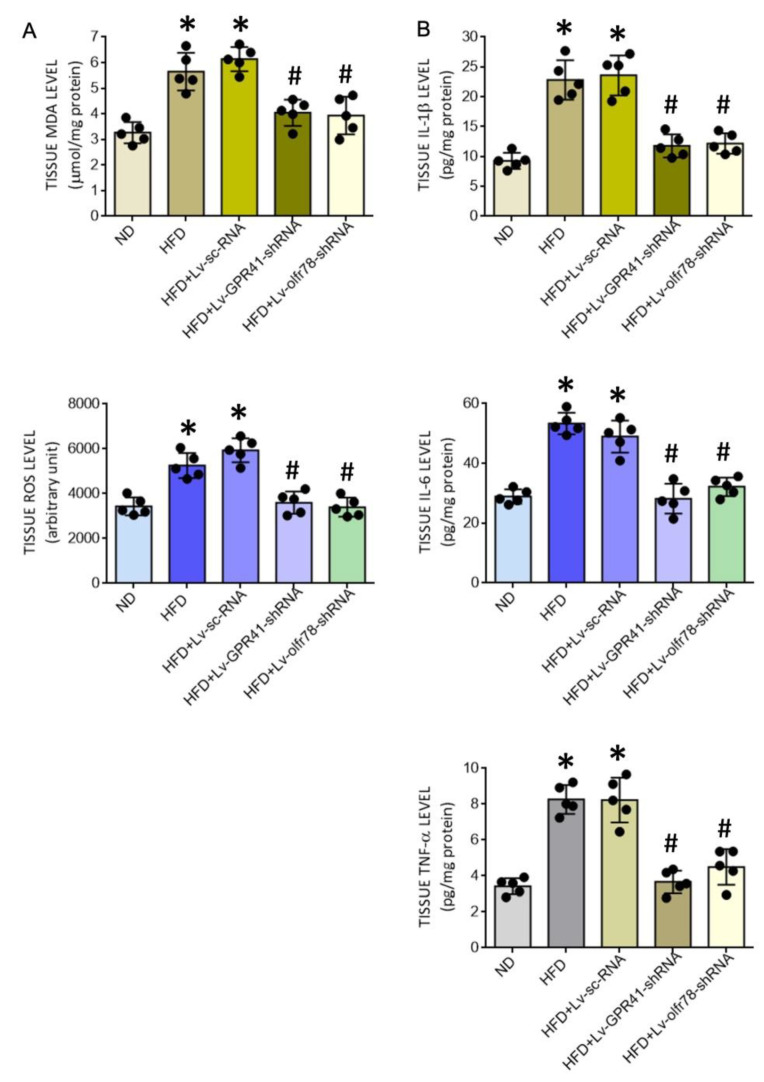
Tissue MDA and ROS levels (**A**) and levels of proinflammatory cytokines IL-1β, IL-6, and TNF-α (**B**) in the PVN of adult ND or HFD offspring, alone or with microinjection bilaterally into the PVN of HFD offspring of Lv-GPR41-shRNA (1 × 10^5^ IFU), Lv-olfr78-shRNA (1 × 10^5^ IFU) or Lv-sc-RNA control, performed at the age of 8 weeks. Data are presented as mean ± SD, *n* = 5 in each group. * *p* < 0.05 vs. ND group, and ^#^
*p* < 0.05 vs. HFD group in the post hoc Tukey’s multiple-range test.

**Figure 8 nutrients-14-04306-f008:**
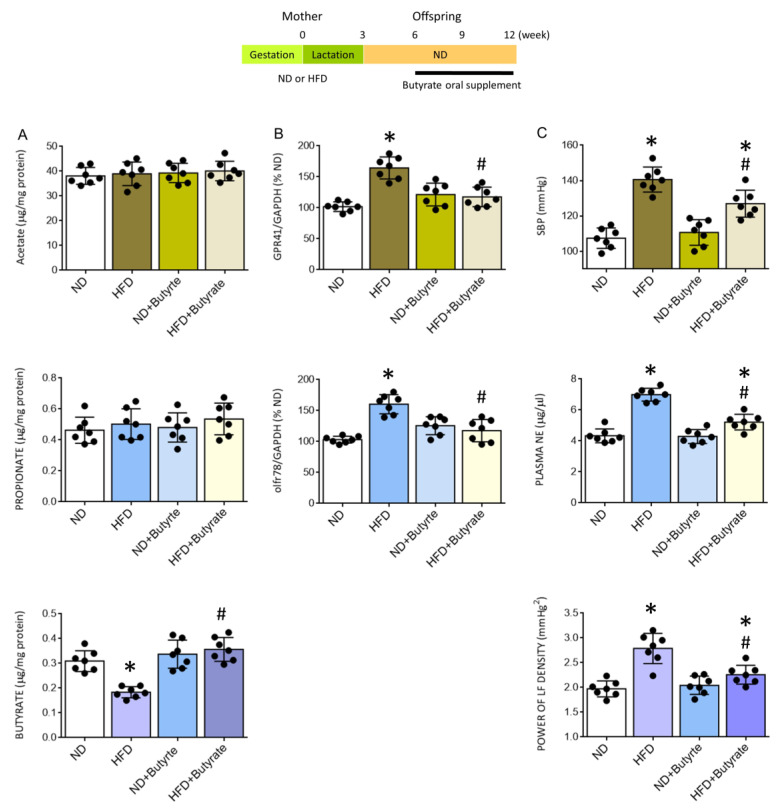
Tissue levels of acetate, propionate, or acetate (**A**), percentage change in the expression of GPR41 or olfr78 mRNA (**B**) in the PVN, and SBP, plasma NE levels, and power density of LF component of SBP (**C**) in adult ND or HFD offspring, alone or with oral supplement in HFD offspring with butyrate (2 g/kg/day) for six weeks. Also presented is the schematic diagram of the experimental design. Data are presented as mean ± SD, *n* = 7 in each group. * *p* < 0.05 vs. ND group, and ^#^
*p* < 0.05 vs. HFD group in the post hoc Tukey’s multiple-range test.

**Table 1 nutrients-14-04306-t001:** Effect of oral supplement with butyrate on markers of oxidative stress and neuroinflammation in the PVN of adult HFD offspring.

	ND		HFD	
	Water Only	+Butyrate	Water Only	+Butyrate
MDA (μmol/mg protein)	3.07 ± 0.32	3.38 ± 0.55	5.20 ± 0.52 *	3.88 ± 0.53 *^,#^
ROS (arbitrary unit)	3263 ± 250	3052 ± 642	4535 ± 515 *	3502 ± 413 ^#^
IL-1β (pg/mg protein)	7.54 ± 1.83	7.95 ± 2.99	19.80 ± 2.44 *	13.16 ± 2.77 *^,^^#^
IL-6 (pg/mg protein)	26.38 ± 6.01	24.42 ± 4.60	51.13 ± 6.89 *	33.29 ± 6.57 *^,^^#^
TNF-α (pg/mg protein)	3.58 ± 0.54	3.19 ± 0.95	8.46 ± 1.55 *	4.39 ± 0.74 ^#^

Butyrate was supplemented in drinking water at 2 g/kg/day for six weeks, beginning at the age of 6 weeks. Data are presented as mean ± SD, *n* = 7 in each group. * *p* < 0.05 vs. ND group, and ^#^
*p* < 0.05 vs. HFD group in the post hoc Tukey’s multiple-range test. MDA, malondialdehyde; ROS, reactive oxygen species; IL-1β, interleukin 1-β; IL-6, interleukin 6; TNF-α, tumor necrosis factor α.

## Data Availability

The data presented in this study are available on reasonable request from the corresponding author and are restricted to investigators based in academic institutions.

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
