# Peer review of "Protection by -Biotics against Hypertension Programmed by Maternal High Fructose Diet: Rectification of Dysregulated Expression of Short-Chain Fatty Acid Receptors in the Hypothalamic Paraventricular Nucleus of Adult Offspring"

_nutrients, 2022, doi:10.3390/nu14204306_

Round 1

Reviewer 1 Report

In this manuscript entitled "Protection by -Biotics of Hypertension Programmed by Maternal High Fructose Diet: Rectification of Dysregulated Expression of Short-Chain Fatty Acid Receptors in the Hypothalamic Paraventricular Nucleus of Adult Offspring", the authors explore the dysregulated tissue levels of SCFAs and expression of SCFA-sensing receptors in the hypothalamic paraventricular nucleus (PVN), a key forebrain region engaged in neural regulation of blood pressure, of offspring to maternal high fructose diet (HFD) exposure. In addition, the authors further investigated the engagement of SCFA-sensing receptors in the PVN on the beneficial effects of -biotics (prebiotic, probiotic, synbiotic and postbiotic) on programmed hypertension primed by maternal HFD. Various oral-biotics improved hypertension of developmental origin by targeting the dysfunctional SCFA-sensing receptors in the PVN to exert antioxidant and anti-inflammatory actions in the brain. Most of data in this MS are convincing and presented by well-designed experiments. The points I was questioning are presented in the limitations of the study, and it is a well-written paper. I recommend that the abstract be presented in < 200 words and that the conclusion be more compact. I hope that my comments are very useful for the improvement of this research.

Author Response

Responses to Reviewer #1

We appreciate very much the affirmative views of the Reviewer on our work, and thank you for the opportunity to improve on our manuscript. Please kindly refer to the pdf file of the revised manuscript for page and line number, as they may be changed in Word file with different version of Microsoft Office of your computer.

I recommend that the abstract be presented in < 200 words and that the conclusion be more compact. I hope that my comments are very useful for the improvement of this research.

Response: Per suggestions by the Reviewer, we have extensively revised the abstract and reduced the number of words in Abstract to < 250. We also moved the second paragraph of the conclusion to P. 19, Lines 766-771 and Lines 790-801 to streamline the Conclusion (p. 19, Lines 803-807 to P. 20, Lines 810-814).

Reviewer 2 Report

The present study was designed to determine whether a dysfunctional gut-brain axis, particularly the microbial metabolites short chain fatty acids (SCFAs) and their receptors in the paraventricular nucleus (PVN), causally underpins hypertension programmed by maternal HFD exposure in adult offspring, and whether SCFA-sensing receptors in the PVN engage in the beneficial effects of -biotics (prebiotic, probiotic, synbiotic and postbiotic) on programmed hypertension primed by maternal HFD. The results indicated that (1) tissue butyrate level was reduced and the expression of SCFA-sensing receptors, GPR41 and olfr78, in the PVN of HFD offspring was increased; (2) these alterations could be rectified by oral supplement with prebiotic, probiotic, and synbiotic; (3) the maternal HFD-primed programmed hypertension in adult offspring was associated with tissue oxidative stress and neuroinflammation in the PVN, which also could be suppressed by oral supplement with the -biotics; (4) silencing GPR41 or olfr78 mRNA in the PVN protected against programmed hypertension, oxidative stress and inflammation in the PVN of adult offspring; (5) oral butyrate supplement restored tissue butyrate level and expressions of GPR41 and olfr78 in the PVN of HFD offspring; and (6) oral butyrate supplement also protected against programmed hypertension in adult HFD offspring. These results together suggest that alterations in tissue butyrate level and expression of the SCFA-sensing receptors, as well as activation of the SCFA-sensing receptor-dependent tissue oxidative stress and neuroinflammation in the PVN, could be novel mechanisms that underlie hypertension programmed by maternal HFD exposure in adult offspring. These findings are interesting and important. This study is well designed and written. However, the following minor revision should be finished before acceptance for publication.

1. In the title, please replace of Hypertension with against Hypertension.

2. Line 15, please replace role with effect.

3. Line 143-146, how were the doses of the -biotics determined? There are any references for it?

4. Line 144, ml should be written as mL. Please check throughout the manuscript for this problem.

5. Line 213, 20,000 X g should be 20,000 × g.

6. Line 339, add against after protects.

7. Line 348, 3X106 should be 3 × 106.

8. Line 362, Tuckys should be Tukeys.

9. In Figure 2A, NDA should be MDA.

Author Response

Responses to Reviewer #2

We appreciate very much the affirmative views of the Reviewer on our work, and thank you for the opportunity to improve on our manuscript. Please kindly refer to the pdf file of the revised manuscript for page and line number, as they may be changed in Word file with different version of Microsoft Office of your computer.

1. In the title, please replace ‘of Hypertension’ with ‘against Hypertension’.

Response: We have replaced “of Hypertension” with “Against Hypertension”.

2. Line 15, please replace ‘role’ with ‘effect’.

Response: Thank the Reviewer for this suggestion. With respect, we submit that the word “role” is more appropriate than “effect” in this sentence (P. 1, Line 15).

3. Line 143-146, how were the doses of the -biotics determined? There are any references for it?

Response: We have edited the text and provided references for the choice of doses of the –biotics (P. 4, Lines 164-166).

4. Line 144, ‘ml’ should be written as ‘mL’. Please check throughout the manuscript for this problem.

Response: We thank the Reviewer for this comment. We have checked throughout the text and made the suggested changes to “mL”.

5. Line 213, ‘20,000 X g’ should be ‘20,000×g’.

Response: Again, we thank the Reviewer for being observant. We have checked throughout the manuscript and made suggested corrections to “×g”.

6. Line 339, add ‘against’ after ‘protects’.

Response: “against” is now added on subtitle 3.1 of the revised manuscript (P. 7, Line 360).

7. Line 348, ‘3X106’ should be ‘3×106’.

Response: We have checked throughout the manuscript and made the suggested corrections.

8. Line 362, ‘Tucky’s’ should be ‘Tukey’s’.

Response: We apologize for this carelessness. We have checked throughout the manuscript and made the suggested corrections.

9. In Figure 2A, ‘NDA’ should be ‘MDA’.

Response: Again, thanks to the Reviewer for being observant. We have made the correction in Figure 2A (on P. 9).
